# Amyloidosis and Longevity: A Lesson from Plants

**DOI:** 10.3390/biology8020043

**Published:** 2019-05-24

**Authors:** Andrei Surguchov, Fatemeh Nouri Emamzadeh, Alexei A. Surguchev

**Affiliations:** 1Department of Neurology, University of Kansas Medical Center, Kansas City, KS 66160, USA; 2Division of Biomedical and Life Sciences, Faculty of Health and Medicine, University of Lancaster, Lancaster LA1 4AY, UK; Fatemeh.NouriEmamzadeh@fda.hhs.gov; 3Section of Otolaryngology, Department of Surgery, Yale School of Medicine, Yale University, New Haven, CT 06520, USA; alexei.surguchev@yale.edu

**Keywords:** amyloidosis, protein aggregation, protein misfolding, longevity, lifespan, synucleins, neurodegeneration

## Abstract

The variety of lifespans of different organisms in nature is amazing. Although it is acknowledged that the longevity is determined by a complex interaction between hereditary and environmental factors, many questions about factors defining lifespan remain open. One of them concerns a wide range of lifespans of different organisms. The reason for the longevity of certain trees, which reaches a thousand years and exceeds the lifespan of most long living vertebrates by a huge margin is also not completely understood. Here we have discussed some distinguishing characteristics of plants, which may explain their remarkable longevity. Among them are the absence (or very low abundance) of intracellular inclusions composed of amyloidogenic proteins, the lack of certain groups of proteins prone to aggregate and form amyloids in animals, and the high level of compounds which inhibit protein aggregation and possess antiaging properties.

## 1. Introduction

### Species Diversity in Maximal Longevity

The longevity of certain vertebrate animals and especially woody trees has long fascinated scientists and the general public. On the other hand, the diversity of lifespans of organisms living on earth is also astonishing. The maximum longevity of different species can vary by 100-fold in mammals and by 1000-fold or sometimes even more if we include in the list the lifespan of invertebrates and mammals [1]. The longevity often correlates with body mass (Figure 1A) [2]. On one end of the spectrum are small and short-lived vertebrates, on the other end are the large and long-lived animals (Figure 1B).

The naked mole rat *Heterocephalus glaber* is the longest living rodent species, which lives 10 times longer than other rodents of comparable size [3,4,5].

We can add to the list of the shortest-lived organisms a vertebrate killifish with a lifespan of about four months and include the bowhead whale to the list of the longest-living mammals at 200 years. Among the invertebrates, bivalve mollusks have a lifespan of an amazing 500 years [10]. According to the database HAGR (The Human Ageing Genomic Resources http://genomics.senescence.info), the medium lifespan of species, classified in agreement with their taxonomic classes, is as follows: Birds- 3–79 years, mammals- 2.1–211, bony fish- 0.16–205, reptilian- 0.4–177, amphibian- 4.1–102, cartilaginous fish- 6–75 years [6]. If we include invertebrates and plants in the list, the amplitude of lifespans will be even greater. Adult life of a rapidly senescing organism – mayfly (insects belonging to the order *Ephemeroptera*) is one or two days [11]. On the other hand, the lifespan of the oldest baobab of Madagascar *Adansonia rubrostipa* (fony baobab, Figure 1C,D), according to the radiocarbon probing of the oldest sample, is 1,136 ± 16 BP [7]. Moreover, the lifespan of some trees calculated on the basis of less sophisticated and accurate methods, such as tree ring counting, may reach 5,150 years [8,9], although these estimates may not be very precise.

## 2. An Overview

### 2.1. Trees Longevity

Trees are the oldest of living organisms on earth. For example, a great basin bristlecone pine nick-named Methuselah (*Pinus longaeva*) living in North America turned 4770 in 2005 and therefore it is currently 4784 years old [12,13]. Several examples of bristlecone pines with an age over 4000 years are also described by Brutovská and coauthors [14]. Although this data may be arguable, reliable results showing an age over 1000 years for many trees are based on an accelerator mass spectrometry radiocarbon measurement. For example, for African baobab Grootboom (*Adansonia digitata* L.) it is estimated to be 1275 ± 50 years, making Grootboom the oldest known angiosperm tree with reliable dating results (Figure 1E) [9,15]. The maximum ages of old trees can be also found in an OLDLIST, a database of old trees: http://www.rmtrr.org/oldlist.htm.

### 2.2. Why Do Vertebrate Animals Not Live as Long as Woody Plants?

Many differences between animal and plant biology may explain a high degree of variation in their lifespan, including a combination of ecological, evolutionary, genetic, biochemical and physiological features. There is no universal definition that fully incorporates the different aspects of aging across all species. Aging of living organisms is due to the accumulation of damages to DNA, proteins and other macromolecules, resulting in deterioration of important biological functions. Aging may be considered as a program that is counterproductive for an individual, but beneficial for biological evolution due to increasing the pressure of natural selection [4,16]. In general, the rate of accumulation of such injuries depending on genes controlling DNA repair and telomere’s length should be relatively similar across organisms. However, certain differences in anatomy, physiology and biochemistry between plants and animals may define the distinction in their lifespan. Some of the biological characteristics that could explain extensive longevity are unique to trees, for example, the retention of stem-cell-like meristematic cells after each growth cycle, the aptitude to restore injured parts, generation of clones, etc., are all unique to trees [17]. Another characteristic of plants that diverge them from animals is the presence of an additional genome located in chloroplasts. Chloroplasts acquired its genome from endosymbiosis of a cyanobacterium around 1.5 billion years ago, after which there was a substantial relocation of genes from the chloroplast to the nucleus [18]. Additionally, there are several biological features that shortens the lifespan of animals and humans, which are discussed below in parts 2.3–2.5.

### 2.3. Amyloid Fibrils and Amyloidosis

One of the features reducing the lifespan of animals and humans is concealed in the properties of a group of amyloidogenic proteins produced in their cells, as well as in cells of bacteria and fungi, but very seldom in plants [19]. Proteins synthesized on ribosomes should fold into defined three-dimensional structures in order to become functionally active. However, some proteins have an intrinsic propensity that convert them from their native functional states into either disordered aggregates or amyloids – a highly ordered insoluble cross-β-sheet fibrils (Figure 2 and Figure 3) [19,20,21].

Amyloid fibrils are formed from monomeric proteins in the course of nucleated polymerization processes generating thermodynamically stable quaternary structures. The propensity of a protein to participate in self-assembly pathways leading to amyloid fibrils is determined by amyloidogenic regions of the protein, which might contain specific amino acid sequences that drive amyloidogenesis [22]. Such amyloids may be deposited as inclusion bodies in various tissues (Figure 4 and Figure 5) [23,24], and their accumulation may lead to conformational diseases or proteopathies [25,26,27].

### 2.4. Amyloidosis in Humans

Proteopathies include Alzheimer’s disease, Parkinson’s disease, Type 2 diabetes, Creutzfeldt–Jakob disease and other disorders. In addition to the gain of toxic properties amyloidogenic proteins can lose their normal function because of the reduction in the intracellular level of their monomeric form, thus aggravating the pathology [28].

In spite of recent advances in the understanding of the amyloid fibrils structure and the mechanisms by which they are formed, there are no efficient approaches to prevent their formation and no effective treatment for conformational diseases.

The ability to self-assemble into ordered amyloid-like β-sheet enriched structures is a common property shared by many polypeptides and proteins not necessarily associated with human diseases [22,25,26,27,29,30,31,32]. At the same time, unstructured protein aggregate formation is a ubiquitous process occurring across the different kingdoms of life. The accumulating data point is the possibility that one of the reasons of plant longevity may be related to the absence of amyloid-like fibrillar inclusions in their cells [33], although plants contain potentially amyloidogenic proteins [32]. The lack or very low level of amyloidogenic inclusions may be explained by the following reasons: **a**) The absence in the plant genome of genes (or gene families) encoding proteins that possess high amyloidogenic propensity in animals and humans (part 2.8) and **b**) the presence of inhibitors of amyloidosis in plant cells (part 2.9).

### 2.5. Amyloidosis in Animals

Amyloidosis has been thoroughly investigated in humans, since it is often associated with human diseases. However, it often occurs in a wide variety of mammals and birds, both domesticated and living in wild nature. For example, amyloidosis has been identified as an important cause of squirrel morbidity and mortality (19.3% of deaths) [34]. It is also described in a brown hare (*Lepus europaeus*) [34] and black-footed wild cat living in South Africa [35]. Amyloidosis is found in association with different chronic diseases in cheetah (*Acinonyx jubatus*), Siberian tigers (*Panthera tigris altaica*) and mink (*Mustela vison*) [36]. Old dogs develop neurodegenerative changes in the brain including cerebrovascular amyloidosis and senile plaques with amyloid deposition, containing Aβ type amyloid, the process similar to Alzheimer’s disease in humans [19,23,24,25,26]. Senile amyloid plaques looking similar to plaques in the Alzheimer’s disease brain of human patients are also common in old non-human primates, including African green *Chlorocebus aethiops* and Cynomolgus monkeys (*Macaca fascicularis*) [37]. Amyloidosis in animals is often associated with various diseases, for example, hepatic or renal failure, significantly shortening their lifespan [36].

### 2.6. Functional Amyloidosis

Aging-dependent formation of amyloids and amyloid-like protein aggregates causatively associates with several neurodegenerative diseases, including Parkinson’s disease, amyotrophic lateral sclerosis and other pathologies. On the other hand, some amyloidogenic proteins called “functional amyloids” formed from natively folded proteins under stringent control may fulfill diverse functional roles as structural modules, components of biofilms, extracellular matrix or even participate in RNA binding and possess enzymatic activity. They may accomplish a protective function against bacteria and viruses [29,30]. The structural advantages of such functional amyloids that allow them to be conserved in the evolution are their higher stability and resistance to proteolysis compared to monomeric forms. Some of functional amyloids possess unique physiochemical properties, which are used for surface coating and other areas of nanobiotechnology [31]. Functional amyloids are generated at any time of a lifespan, whereas pathological amyloids usually begin accumulating in individuals over their reproductive age and, therefore, there is low selective pressure for their elimination or modification.

Amyloidogenic properties and the ability to self-assemble were recently discovered in proteins for which these features had been difficult to suspect, expanding the group of functional amyloids. For example, Drosophila RNA binding protein Otu (Drosophila ovarian tumor) can form solid amyloid fibers in the presence of RNA [38]. Formation of amyloid fiber is directed by prion-like repeats in the intrinsically disordered C-terminal Otu domain. Remarkably, Otu possesses deubiquitinase activity, which dramatically increases as a result of protein polymerization and amyloid formation and is regulated by RNA binding. Furthermore, Otu controls excessive inflammation, delaying the aging process, and mutations in Otu shorten the longevity of Drosophila, indicating that Otu plays an important role in the extension of Drosophila lifespan [38]. Thus, some functional amyloids may play a beneficial role fulfilling important functions in an organism.

### 2.7. Rare Amyloidosis in Plants 

The formation of amyloid fibrils occurs not only in humans and animals, but also in fungi and bacteria, however, this process is very rare in plants. The authors of several publications assume that amyloid properties have not been shown under native conditions for any plant protein, although many potentially amyloidogenic proteins are present in plants [39]. The bioinformatic analysis performed using several algorithms, for example, prediction algorithms TANGO, Waltz and SARP (Sequence Analysis based on the Ranking of Probabilities) revealed that potentially amyloidogenic proteins were abundant in the proteomes of many land plants. However, in spite of the susceptibility of plant proteomes to protein aggregation, insoluble amyloid fibrils are very rare in plants [39]. It should be noted that some plant amyloidogenic proteins possess protective properties thus contributing to the prolongation of the lifespan. For example, defensins from the radish *Raphanus sativus* exhibit fungicidal activity [40]. Garvey and coauthors (2013) [40] examined an antifungal protein RsAFP-19 and highly amyloidogenic fragment of this protein from *Raphanus sativus*. Interestingly, a fibril-forming capacity of this amyloid was easily manipulated by externally controlled conditions, for example, by freezing and thawing [40].

Another example of the protective effect of plant amyloidogenic peptide possessing antimicrobial properties is Cn-AMP2. This 11-amino acid peptide is synthesized in the liquid endosperm of coconut, *Cocos nucifera* [41]. Cn-AMP2 possesses amyloidogenic propensity comparable with that of β-amyloid from Alzheimer’s disease plaques. In vitro Cn-AMP2 easily aggregates forming fibrillar structures with typical Congo red absorbance spectra, distinctive thioflavin T fluorescence and fibrillar morphology under TEM [41].

Several authors describe rare cases of amyloidosis in plants, which occurs in specific, rather exotic cases and sites, when these proteins are used as functional amyloids to fulfil a specialized function. For example, proteinaceous, pleated β-sheet highly ordered complexes are present in extracellular polymeric substances of terrestrial alga *Prasiola linearis* [42,43]. These amyloid-like structures play the role of a glue for this multicellular green alga and participate in a generic mechanism ensuring mechanical strength in natural algal adhesives. The amyloid features of these structures were confirmed by a green-gold birefringence in cross-polarized light after Congo red staining, Raman spectroscopy, chemical staining, and atomic force microscopy. The structural properties of such amyloids explain an easy attachment of these microalgae to various surfaces in the urban environment [43].

Another example of a highly specialized amyloidogenic protein in plants is the elongation factor REF or Hevb1—a major component of latex in the “rubber tree” *Hevea brasiliensis.* REF participates in natural rubber biosynthesis, contains β-sheet organized aggregates with amyloid properties proven by circular dichroism (CD), TEM, infra-red spectroscopy and X-ray diffraction [44]. Hevea is the genus of flowering plants in the spurge family used commercially for rubber production.

Occasionally the formation of amyloidogenic inclusions in plants is not due to authentic fibrillation of endogenous proteins but has been detected in genetically modified plants by producing heterologous proteins or by plant pathogenic bacteria. For example, Oh and coauthors (2007) [45] found spherical oligomers, protofibrils, and β-sheet-rich fibrils in tobacco leaves composed from the heat-stable, glycine-rich type III-secreted proteins called hairpins. These proteins were produced by gram-negative plant pathogenic proteobacteria from the genus *Xanthomonas*. The fibrillar form of one of these proteins—His6-HpaG behaves as a typical amyloid protein, the fibrillation of which is modulated by an amino acid motif in the C-terminus of the protein [45]. Another example of fibril formation by amyloidogenic proteins was described by Villar-Pique et al. [46]. The authors investigated amyloidosis of maize transglutaminase in the chloroplasts of tobacco transplastomic plants. A transplastomic plant is a genetically modified organism in which new genes have not been inserted in the nuclear DNA but are introduced into the DNA of the chloroplasts. The fact that inclusion bodies containing maize transglutaminase have an amyloid-like nature was proven by FTIR absorbance, absorption spectra of Congo red by birefringence, characteristic amyloid birefringence in cross-polarized light and fluorescence emission spectra of thioflavin T [46]. Thus, these amyloids share some structural features with the inclusions consisting of Aβ-peptide, α-synuclein, and prion proteins described in human neurodegenerative diseases. Therefore, the overproduction and accumulation of misfolded proteins after their translation on plant ribosomes may cause their self-assembly into ordered β-sheet amyloid structures. However, as a rule this mechanism is used in plants in rare cases in order to fulfill a highly specialized function often associated with cell protection. In order to monitor the transition from a primarily monomeric peptide into fibrils the analysis is usually conducted in multiple wells with a subsequent reading in a microplate reader. In addition to the method of amyloidosis monitoring that was aforementioned, a real-time fibrillization assay could also be used based on a fluorescence or UV–vis spectrometer in modified NMR tubes [47]. In addition to the method of amyloidosis previously mentioned, electrochemical techniques can be applied, which give additional information about amyloid formation [48]. In addition, luminescent complexes were used for monitoring amyloidosis [49].

### 2.8. Certain Amyloidogenic Proteins Highly Expressed in Vertebrates are Absent in Plants—An Example is the Synuclein Family

Some amyloidogenic proteins highly expressed in animals and humans, such as synucleins, are absent in plants. Synucleins are a family of three small naturally unfolded proteins (α-, β- and γ- synucleins) predominantly expressed in neural tissues. The most thoroughly studied isoform is α- synuclein associated with Parkinson’s disease, dementia with Lewy bodies and multiple system atrophy [50]. α- Synuclein is highly expressed in neural tissues and its amount in the brain may reach up to 1% [51]. Later studies demonstrated that all members of the family were also present in other organs and tissues [50]. α-Synuclein is a major protein component of Lewy bodies—a hallmark of Parkinson’s disease and is also found in the brain of patients with dementia. Importantly, synucleins are expressed only in vertebrates, and no ortholog or homolog proteins with similar amino acid sequences are detected in the genome of other organisms, including plants. In vertebrates the synuclein sequences are highly conserved across species, suggesting functional constraints. We can speculate that a precursor of a gene encoding synuclein appeared relatively late in eukaryotic evolution, evolved de novo due to mutations in a noncoding DNA sequence and then gave rise to two other isoforms as a result of duplication and divergence leading to a family of three conservative homologs (α-, β-, and γ-synucleins) in vertebrate species. The unique conservation of the synucleins sequences is consistent with an important physiological role for these proteins, for example, in synaptic functions. In this case, lifelong positive selection pressure mediates conservation and persistence of synucleins in the vertebrate genome.

As many de novo formed genes [52,53], synucleins are relatively short (127–140 amino acids) and inherently unstructured, adopting a folded shape only on contact with membranes or binding partners.

Their appearance may be explained by the evolutional formation of the brain and CNS, which has required the emergence of new proteins fulfilling synaptic functions. Recent fast progress in whole genome and large-scale sequencing provides much evidence that new genes have evolved and keep appearing from noncoding parts of the genome [52,53,54].

Since synucleins are highly expressed in vertebrates, but their genes are absent in plants, including long-lived plants, we questioned what the structure was for an organism with an unusually high lifespan, i.e., the longest living rodent naked mole rat [2,3,4,5]. Interestingly, amino acid sequences of synucleins in naked mole rat are significantly different compared to the sequences of synucleins from other vertebrates. In particular, two members of the family possessing amyloidogenic properties [23,27,50,55] α- and γ-synucleins had substitutions of those amino acids, which determined high amyloidogenic properties of these proteins in human and other mammalian species (Figure 6).

For example, serine-87 (S^87^) in human α-synuclein, which controls its oligomerization via phosphorylation [56] was replaced by asparagine (N) in the naked mole rat α-synuclein. Phosphorylated Ser87 was considered a pathological hallmark of α-synuclein inclusions [53,54,55]. Furthermore, phosphorylation of human α-synuclein at residues S87 induced the formation of extended protein conformations exposing the aggregation prone α-synuclein NAC region thus increasing the aggregation propensity [56,57,58]. Therefore, substitution of S87 by asparagine in naked mole rat α-synuclein should reduce the propensity to aggregation and formation of amyloid. Another amyloidogenic protein- γ-synuclein from the naked mole rat contained nine amino acids which were not found in the corresponding positions in other vertebrates. The central part of γ-synuclein contained four unique aliphatic alanines, which made this part of the protein very hydrophobic (Figure 6B). We can only assume based on the amino acid properties that the alterations in naked mole rat synucleins compared to other mammalian species reduced their amyloidogenic properties, since this effect was not examined experimentally. However, similar analysis was done for another aggregation-prone polypeptide, amyloid-beta (Aβ). Aβ from naked molar rat differed from humans by only one amino acid at position 13. The human Aβ contained histidine, while in the naked molar rat this position was occupied by arginine. However, due to this slight amino acid substitution (H13R) in Aβ from the naked molar rat the polypeptide had a significantly lower propensity to aggregate (10.76 ± 0.33 A.U.) than the human form (14.33 ± 0.47 A.U., *p* = 0.011) [59]. Thus, the comparison of amino acid sequences of several amyloidogenic proteins from the naked mole rat and other mammals allows for the suggestion that amino acids located in key positions determining the predisposition to amyloidogenesis were replaced in the naked mole rat proteins by amino acids which did not promote amyloidosis.

### 2.9. Plant Longevity and Secondary Metabolites Inhibiting Amyloidosis and Preventing Aging

The fact that amyloidogenic inclusions in plants are absent or at least very rare compared to those found in mammals may be explained by a high concentration of compounds inhibiting protein aggregation in plant cells [60]. Recently it was demonstrated that protein extract from the sugar maple tree *Acer saccharum* easily fibrillated without low molecular substances in the extract, but fibrillation was strongly inhibited by the addition of polyphenols and other small phytomolecules from the same plant [33]. These two facts: very low amounts or the absence of amyloids and the high content of inhibitors of amyloidogenesis, including polyphenols and antioxidants, might explain high plant longevity. Furthermore, a correlation between plant longevity and the ability to inhibit protein aggregation has been described [33]. Importantly, many inhibitors of amyloidosis possess antiaging properties [60,61,62].

Testing of plant phytomolecules, such as alkaloids, phenols, steroids, etc. in model organisms has demonstrated that many of them have a dual effect, possessing simultaneously anti-aggregational and antiaging properties [60,61,62]. One example of such a substance is tambulin—a hydroxy substituted flavanol from fruits of *Zanthoxyllum armatum DC*. The effect of tambulin was examined in nematode *Caenorhabditis elegans* (C. elegans) and found to be longevity promoting (16,79% by 50 µM tambulin) and possessing neuroprotective and neuromodulatory activities [62]. Furthermore, tambulin inhibits α-synuclein aggregation in *C. elegans*. Interestingly, tambulin upregulates mRNA expression of ROS scavenging genes and genes associated with longevity, for example, daf-16, sod-1, sod-3, and ctl-2, as a result significantly enhancing lifespan and stress tolerance of *C. elegans* [62].

Antiaging and anti-aggregational properties are described for many plant substances. For example, polyphenol resveratrol (3,5,4′-trihydroxystilbene) isolated from the root of *Veratrum grandiflorum* and present in high amounts in many fruits, such as blueberries (*Vaccinium spp*.), blackberries (*Morus spp.),* grapes (*Vitaceae*) and peanuts (*Arachis hypogaea*) possesses these properties. Resveratrol extends lifespan in humans and all model organisms, including mammals, the budding yeast *Saccharomyces cerevisiae*, *C. elegans* and *Drosophila melanogaster* [61].

The activity of many substances from plants extend longevity considerably and prevent or delay the beginning of age-related disorders in many experiments with model systems and regulating aging-associated pathways. Antiaging and anti-aggregational compounds isolated from plants stimulate autophagy and DNA repair, and simultaneously inhibit the deleterious effects of reactive oxygen species, preventing oxidation [63,64,65,66]. Thus, plants contain antiaging molecules that prolong lifespan in experiments with model organisms and affect different aging-related pathways, one of which is amyloidosis. It is natural to assume that the same plant compounds possess a similar effect in their own cells.

Interestingly, the results of high-throughput nano probes-based screening for protein aggregation inhibitors demonstrated that perennial plants contained more effective compounds compared to shorter-lived annual plants, supporting the idea about the association of protein aggregation inhibitors with plant longevity [33,67].

The anti-inflammatory, antioxidant and anti-aging effects of phytomolecules are documented in many studies. However, some aspects of their action are still far from being understood. For example, the examination of their role in epigenetic regulation of gene expression is at the initial stage and requires further analysis [65]. An interesting example of the epigenetic effect of a phytomolecule—isothiocyanate sulforaphane from cruciferous was recently described. Sulforaphane activates the antioxidant and anti-inflammatory responses and modifies mitochondrial dynamics by inducing the Nrf2 pathway and inhibiting NF-κB. In addition, sulforaphane has an epigenetic effect by inhibiting histone deacytelases (HDACs) and DNA methyltransferases [68,69], thus regulating the expression of specific genes.

## 3. Conclusions

Here we discussed the hypothesis that high plant longevity was due to a combination of several factors, including a low level of amyloidosis in plants cells, the absence of certain genes encoding amyloidogenic proteins in the plant genome, and a high level of phytomolecules inhibiting the formation of amyloids. These phytomolecules are just one means of remedy and defense against amyloidosis in plants. The other includes the organization of plant cell walls preventing the propagation of amyloidogenic proteins between cells. Amyloidogenic proteins associated with a shorter lifespan in vertebrates and humans may easily spread between different organs and tissues due to a prion-like mechanism of propagation and relatively thin membrane separating cells. In those rare cases when plants produce amyloidogenic proteins for a special function, the thick multilayer cell wall built of cellulose, hemicellulose, pectin and other substances serves as a physical barrier, which impedes an easy propagation of proteins between cells. The effect of other means of plant defense preventing amyloidosis and its propagation was not investigated in detail. This may include epigenetic regulation of gene expression by phytomolecules [66], the effect of microRNA on lifespan [70,71] and the role of a unique set of plant small heat shock proteins (sHSPs) on amyloidosis and longevity [72].

## Figures and Tables

**Figure 1 biology-08-00043-f001:**
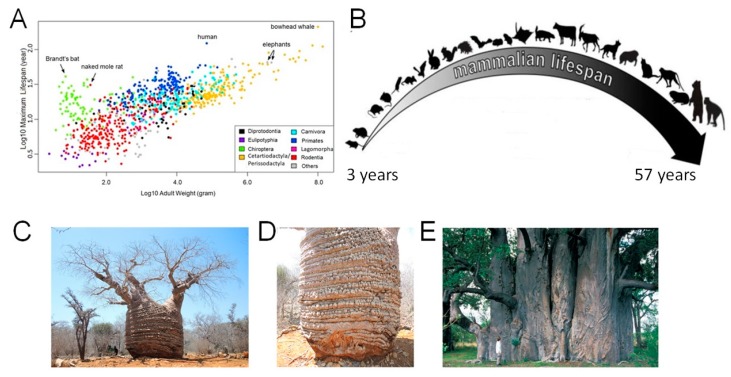
Lifespan variation across different species. (**A**) There is a positive correlation between the lifespan and weight of adult mammals [2]. Data for adult body mass and maximum lifespan records of mammalian species are from an age database [6]. (**B**) Shrews such as the *Suncus etruscus* is a small and short-lived species which weighs approximately two grams and has a lifespan of nearly 3.2 years. The other extreme are the large and long-lived animals: African elephant (*Loxodonta africana*) is the largest land mammal with an average weight of six tons and a lifespan of seventy years. The lifespan of plants is considerably longer than animals. The ages of some trees is well over 1,000 years. (**C**) The general view of the oldest baobab of Madagascar *Adansonia rubrostipa* (fony baobab). (**D)** Tri-stemmed trunk of the Grandmothers baobab with bulbous formations. (**E**) The southern flank of Grootboom, the largest known African baobab. Early African explorers attempted to extrapolate the low growth rate of old baobabs over their entire lifecycle, claiming their age to be up to 5,150 years for the largest individuals [7,8]. A,B—modified from [2]; C,D—from [7], E—from [9].

**Figure 2 biology-08-00043-f002:**
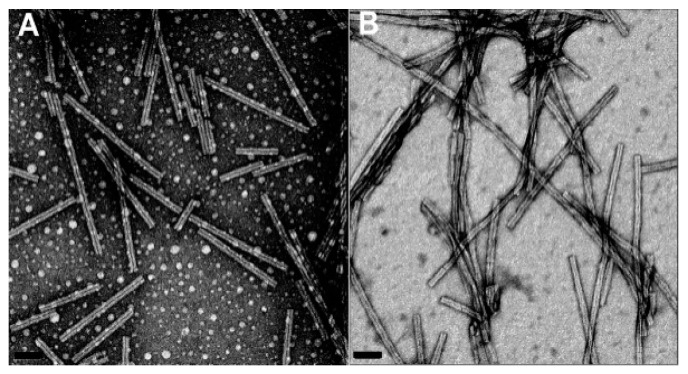
Negatively stained images of the protein dsGB1—an amyloid-forming mutant of the immuno-globulin binding domain of streptococcal protein G used as a model for protein folding and stability studies. Circular assemblies as well as fibrils are visible on **A**, and mature dsGB1 fibrils having a homogeneous and periodically twisted morphology are seen on **B**. Transmission electron microscopy (TEM), scale bar 50 nm. Modified from [20].

**Figure 3 biology-08-00043-f003:**
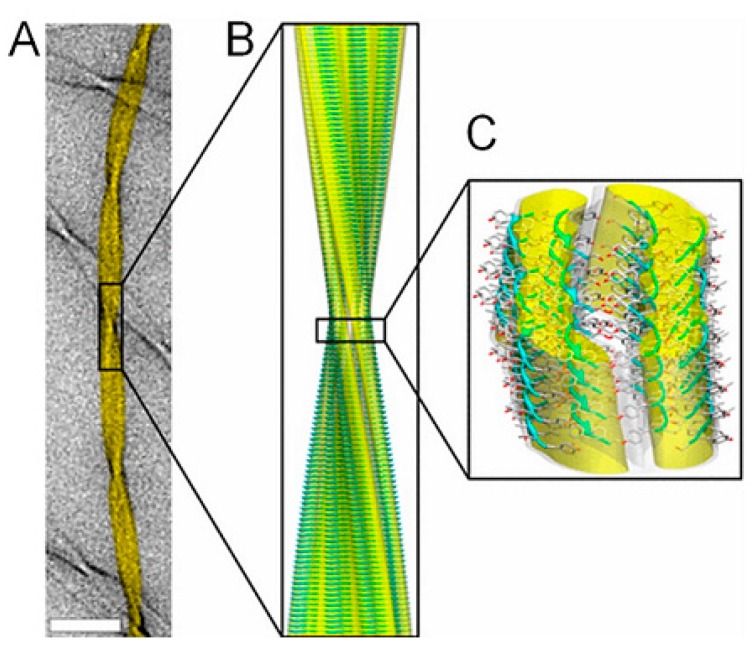
Atomic resolution structure (0.5 Å) of cross-β-amyloid fibrils formed by an 11-residue fragment of the protein transthyretin. The fibrils have the classic amyloid morphology and are 100–200 Å in diameter and 1–3 μm in length. Highly ordered cross-β-core structures are composed of arrays of ß-sheets running parallel to the long axis of the fibrils. (**A**) The background image of the fibril taken using TEM (scale bar—50 nm). (**B**) Magic-angle spinning nuclear magnetic resonance (MAS NMR) atomic-resolution structure of the triplet fibril fitted into the cryo-EM reconstruction. (**C**) The fibril surfaces with the constituent ß-sheets in a ribbon representation. Oxygen-red, carbon-gray, nitrogen-blue. The images were reconstructed using a combination of several biophysical techniques: X-ray fiber diffraction, cryoelectron microscopy, scanning TEM and atomic force microscopy. Other details are provided in [21].

**Figure 4 biology-08-00043-f004:**
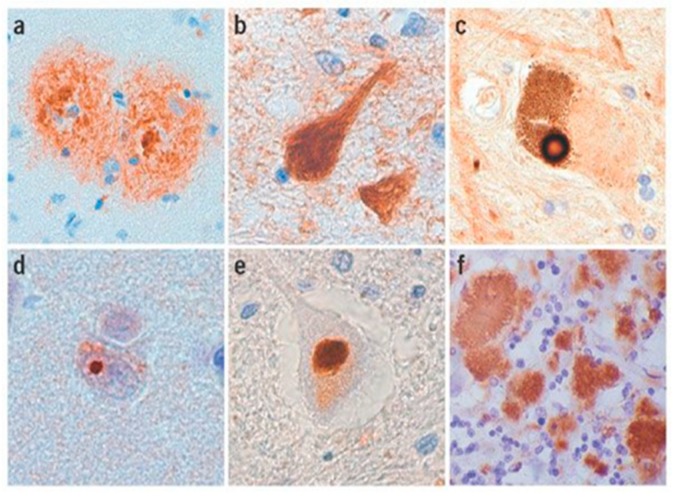
(**a**) Senile plaques in the neocortex of an Alzheimer’s disease patient. (**b**) Neuro-fibrillary tangles (NFTs) in the hippocampus of a patient with frontotemporal dementia with parkinsonism-17 (FTDP-17) (R406W mutation). (**c**) Lewy body in the substantia nigra of a Parkinson’s disease patient (**d**) Intranuclear polyglutamine inclusion in the neocortex of a Huntington disease patient. (**e**) Ubiquitinylated inclusion in the spinal cord motor neuron of an amyotrophic latera; sclerosis (ALS) patient. (**f**) Protease-resistant prion protein (PrP) in the cerebellum of a frontotemporal dementia with parkinsonism-17 patient [23].

**Figure 5 biology-08-00043-f005:**
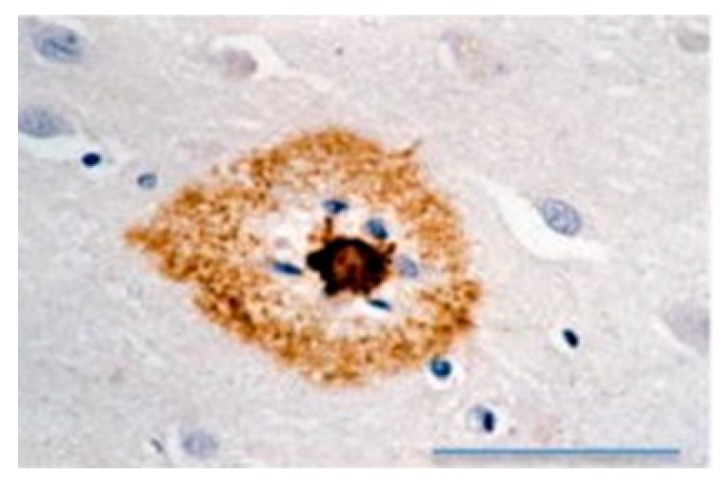
Amyloid-beta (Aβ) immunostaining (brown) of an amyloid plaque with a core in a human Alzheimer’s disease case. The dense dark brown core and blue glial nuclei are encircled by a halo of diffuse Aβ. Scale bars represent 50 μm [24].

**Figure 6 biology-08-00043-f006:**
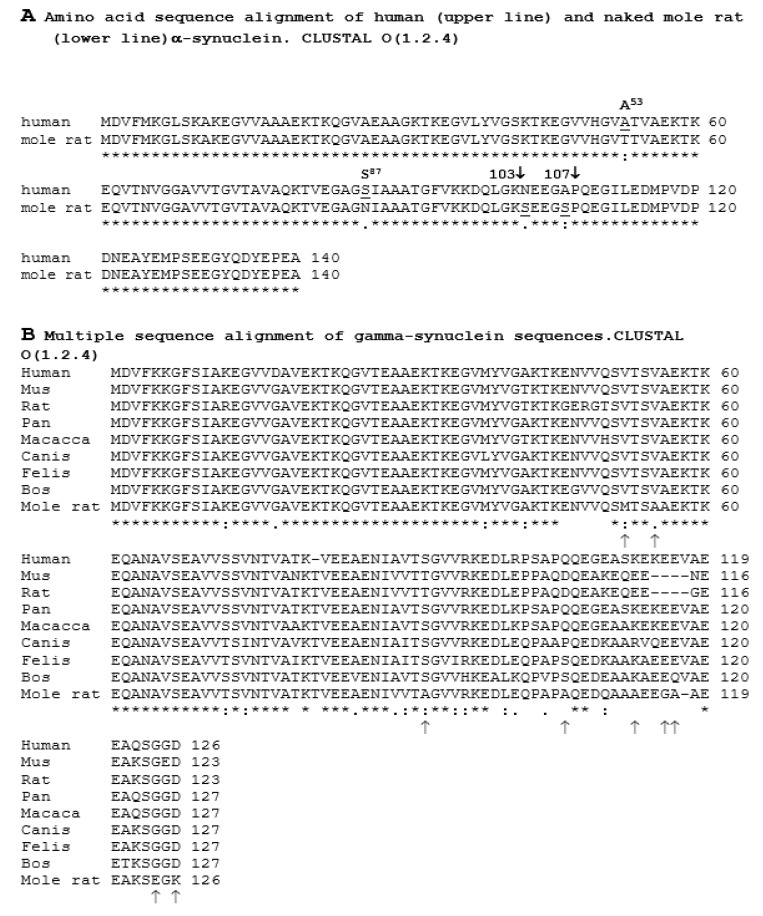
Alignment of amino acid sequences of synucleins. (**A**) N-termini of α-synuclein from a human and naked mole rat have high similarity in amino acid sequences. However, the localization of critical serine residues in the central part of the protein molecule is different. Phosphorylation of Serine^87^ in the human protein regulates α-synuclein oligomerization, changes amyloidogenic properties of the protein and influences synuclein-membrane interactions [56]. On the other hand, serine residues in positions 103 and 107 of α-synuclein from naked mole rat are located in the region which participates in the long-range interactions with the hydrophobic NAC region (amino acids 61–95) determining the ability of α-synuclein to aggregate and form amyloids [57]. (**B**) Alignment of γ-synuclein amino acid sequences from vertebrates. N-termini have very conservative sequences, whereas C-termini are more variable. γ-Synuclein from naked mole rat contained nine unique amino acids, including five alanine residues located predominantly in the C-terminus of the protein. Arrows at the bottom show unique amino acids in the naked mole rat protein. The alignment was performed using CLUSTAL O(1.2.4).

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
