# Peer review of "Amyloidosis and Longevity: A Lesson from Plants"

_biology, 2019, doi:10.3390/biology8020043_

Round 1

Reviewer 1 Report

I highly recommended accepting this paper for publishing in a wide profile journal, as the  “BIOLOGY”. The phenomenon of the difference in the livespane of different species of living organisms has not yet found yet a satisfactory explanation, despite the efforts of many researchers. Therefore, the introduction of new fact-based hypotheses should be welcomed. Such one is presented in the paper of Surguchev et al.

Illustratistions in Figures are perfect!

Author Response

Thank you very much for reviewing of our manuscript and high opinion about our work.

Reviewer 2 Report

The authors Andrei Surguchov, Fatemeh Nouri Emamzadeh and Alexei Surguchov have submitted a manuscript (ID: biology-503943, type: Opinion) entitled “Amyloidosis and longevity: a lesson from plants” to the section “Biochemistry and Molecular Biology” of MDPI Biology. Thee authors discuss an extremely interesting topic in a scientific solid and competent way. Indeed, some distinguishing characteristics of plants which are related amyloidogenic proteins may explain their remarkable longevity. In order to improve this manuscript additional biophysical methods and techniques which are suited to study amyloid fibrillization should be mentioned and explained. The corresponding analytical tools are described in actualpublications showing results about amyloid fibrillizationobtained by techniques such as NMR spectroscopy,AtomicForceMicroscopy(AFM)in combination with Thioflavin T (ThT) fluorescenceassays, and in silico/ docking studies.

In summary: the manuscript of the authors is a valuable and helpful contribution to topic under study and would fit perfectly in the section “Biochemistry and Molecular Biology” of MDPI Biology.

Author Response

Dear Reviewer 2:

Thank you for reviewing our manuscript and for your suggestions. In respond to your suggestion we added the following text and references:

In order to monitor the transition from a primarily monomeric peptide into fibrils the analysis is usually conducted in multiple wells with a subsequent reading in a microplate reader. In addition to the method of amyloidosis monitoring that we already mentioned above, a real-time fibrillization assay can be also used based on a fluorescence or UV–vis spectrometer in modified NMR tubes (50 Cook and Marti, 2012). In addition to the method of amyloidosis that we already mentioned above electrochemical techniques can be applied which give an additional information about amyloid formation (51 Veloso et al., 2009). In addition luminescent complexes are used for monitoring amyloidosis (52 Cook et al., 2011).

 50 Cook, N.P.; Martí, A.A. Facile methodology for monitoring amyloid-β fibrillization. ACS Chem Neurosci. 2012, 3 (11):896-9. doi: 10.1021/cn300135n

51 Veloso, A.J.; Hung, V.W.; Sindhu, G.; Constantinof, A.; Kerman, K. Electrochemical oxidation of benzothiazole dyes for monitoring amyloid formation related to the Alzheimer's disease. Anal Chem. 2009, 81(22):9410-5. doi: 10.1021/ac901940a.

52 Cook, N.P.; Torres, V.; Jain, D.; Martí, A.A. Sensing amyloid-β aggregation using luminescent dipyridophenazine ruthenium(II) complexes. J Am Chem Soc. 2011, 133 (29):11121-3. doi: 10.1021/ja204656r.

We also changes the numeration of references after Ref. 50 correspondingly.

Thanks again

Reviewer 3 Report

An original point of view about amyloids from plants to animals, a state of the art review !

Author Response

Thank you very much for the review and for the high opinion about our manuscript